# Molecular Manipulation of the miR399/*PHO2* Expression Module Alters the Salt Stress Response of *Arabidopsis thaliana*

**DOI:** 10.3390/plants10010073

**Published:** 2020-12-31

**Authors:** Joseph L. Pegler, Jackson M.J. Oultram, Christopher P.L. Grof, Andrew L. Eamens

**Affiliations:** Centre for Plant Science, School of Environmental and Life Sciences, Faculty of Science, University of Newcastle, Callaghan, NSW 2308, Australia; joseph.pegler@newcastle.edu.au (J.L.P.); Jackson.Oultram@uon.edu.au (J.M.J.O.); chris.grof@newcastle.edu.au (C.P.L.G.)

**Keywords:** *Arabidopsis thaliana*, molecular manipulation, microRNA399 (miR399), *PHOSPHATE2* (*PHO2*) gene expression regulation, salt stress, RT-qPCR

## Abstract

In *Arabidopsis thaliana* (*Arabidopsis*), the microRNA399 (miR399)/*PHOSPHATE2* (*PHO2*) expression module is central to the response of *Arabidopsis* to phosphate (PO_4_) stress. In addition, miR399 has been demonstrated to also alter in abundance in response to salt stress. We therefore used a molecular modification approach to alter miR399 abundance to investigate the requirement of altered miR399 abundance in *Arabidopsis* in response to salt stress. The generated transformant lines, *MIM399* and *MIR399* plants, with reduced and elevated miR399 abundance respectively, displayed differences in their phenotypic and physiological response to those of wild-type *Arabidopsis* (Col-0) plants following exposure to a 7-day period of salt stress. However, at the molecular level, elevated miR399 abundance, and therefore, altered *PHO2* target gene expression in salt-stressed Col-0, *MIM399* and *MIR399* plants, resulted in significant changes to the expression level of the two PO_4_ transporter genes, *PHOSPHATE TRANSPORTER1;4* (*PHT1;4*) and *PHT1;9*. Elevated PHT1;4 and PHT1;9 PO_4_ transporter levels in salt stressed *Arabidopsis* would enhance PO_4_ translocation from the root to the shoot tissue which would supply additional levels of this precious cellular resource that could be utilized by the aerial tissues of salt stressed *Arabidopsis* to either maintain essential biological processes or to mount an adaptive response to salt stress.

## 1. Introduction

In recent decades, soil salinization has become a major environmental concern with approximately 20% of the world’s land usable for agriculture now adversely affected by saline soils [1,2]. Excessive salt accumulation in the soils of arable land can result from natural causes such as coastal flooding events and/or elevated evapotranspiration rates. However, more recently, anthropogenic activities such as poor irrigation practices, excessive fertilizer application, over mechanical cultivation, and/or a lack of appropriate vegetative ground cover, have greatly increased both the rate and degree of accumulation of salt in the soils of once productive agricultural land [3,4,5]. The standard growth and development, and therefore yield potential, of each of the major crop species, including wheat (*Triticum aestivum*), maize (*Zea mays*) and rice (*Oryza sativa*), is negatively impacted by elevated levels of salt in the soil [6,7,8,9,10]. Once taken up by the root system, the salt ions are further concentrated in the aerial tissues of a plant to interfere with numerous molecular, biochemical and physiological processes [4,5,6,9]. Moreover, a high abundance of salt ions in the aerial tissues not only leads to ionic toxicity, but can also enhance osmotic pressure, reduce turgor pressure, and promote the generation of reactive oxygen species (ROS), with elevated levels of ROS leading to the onset of oxidative stress [11,12,13]. Together, this promotes the closure of leaf stomata, reduces the rate of photosynthesis, decreases the uptake of nutrients and water from the soil, and represses carbon assimilation, all of which in turn result in the termination of cell elongation and expansion, while simultaneously promoting the premature transition from vegetative to reproductive development and the early onset of senescence [5,9,14,15].

Plants are restricted to their site of germination, and therefore, have developed highly intricate and interrelated molecular pathways which are essential to their ability to rapidly adapt to changes to the surrounding environment; adaptive measures initiated by a plant to attempt to restrict the degree to which its developmental progression is compromised [9,16,17,18]. At the molecular level, specifically the posttranscriptional level, the microRNA (miRNA) class of small regulatory RNA (sRNA) has been repeatedly demonstrated to direct a central role in coordinating the gene expression changes required by a plant to mount an adaptive response to alterations to its growth environment [19,20,21,22,23]. Plant miRNAs are predominantly 21 nucleotides (nt) in length and are processed from imperfectly double-stranded RNA (dsRNA) stem-loop structured precursor transcripts in the plant cell nucleus by the DOUBLE-STRANDED RNA BINDING1 (DRB1) and DICER-LIKE1 (DCL1) protein partnership [24,25,26,27]. Post precursor transcript processing by DRB1/DCL1, the now mature miRNA is loaded into the RNA-induced silencing complex (RISC), a protein complex that contains the endonuclease ARGONAUTE1 (AGO1) at its catalytic core (reviewed in [28]). RISC uses the loaded miRNA as a sequence specificity guide to direct expression regulation of target gene transcripts that harbor highly complementary miRNA target site sequences [29]. Due to the high level of miRNA/target gene sequence complementarity, plant miRNAs were originally thought to only regulate the expression of their target genes via a messenger RNA (mRNA) cleavage mechanism of RNA silencing, more recently however, translational repression has also been identified as an RNA silencing mechanism directed by plant miRNAs [28,29,30].

The miRNA, miR399, was originally identified in *Arabidopsis thaliana* (*Arabidopsis*) and has since been demonstrated to be central to the ability of a number of plant species to adapt to growth in a phosphorus (P) deplete environment [31,32,33,34]. When P levels are inadequate in the growth environment of *Arabidopsis*, the concentration of phosphate (PO_4_), the primary form of inorganic P (Pi) taken up by the root system is also decreased, and this in turn promotes *MIR399* gene expression to enhance the abundance of miR399 [31,32]. Post loading to AGO1-catalyzed RISC, miR399 directs the posttranscriptional silencing of its target gene transcript, *PHOSPHATE2* (*PHO2*), a transcript that encodes for a ubiquitin conjugation enzyme24 (UBC24) protein [35,36]. The PHO2 UBC24 is proposed to direct ubiquitin-mediated degradation of the PO_4_ transporter proteins, PHOSPHATE TRANSPORTER1;4 (PHT1;4), PHT1;8 and PHT1;9, three PO_4_ transporters required for P translocation from the root to the shoot tissues of *Arabidopsis* [35,37]. Therefore, elevated miR399 abundance in the absence of P, leads to enhanced repression of *PHO2* gene expression, which in turn reduces PHO2 protein abundance, thereby leading to the relaxation of PHO2-directed suppression of PHT1;4, PHT1;8 and PHT1;9 function [35,36,37,38]. Enhanced PHT1 PO_4_ transporter activity, due to the removal of their PHO2-directed repression, leads to enhanced P translocation from the roots to the shoots in an attempt by *Arabidopsis* to maintain P homeostasis [35,37].

Considering the well documented and crucial role played by the miR399/*PHO2* expression module as part of the molecular response of *Arabidopsis* to attempt to maintain P homeostasis in a P deplete growth environment, it was somewhat surprising to observe that the abundance of miR399 was also responsive to drought and salt stress [19]. Here, we therefore applied a molecular modification approach to alter miR399 abundance via the deployment of endogenous target mimicry knockdown and miRNA precursor transcript over-expression technologies to generate *Arabidopsis* transformant lines with reduced and elevated miR399 abundance, respectively. The resulting transformant lines, termed *MIM399* and *MIR399* plants respectively, were developed to document their phenotypic and physiological response to salt stress for comparison to the response of unmodified wild-type *Arabidopsis* plants. The phenotypic and physiological assessments reported here show that each of the three assessed *Arabidopsis* plant lines displayed a unique response to salt stress. Further, salt-stressed Col-0, *MIM399* and *MIR399* plants additionally displayed a unique response at the molecular level with the abundance of the central miR399/*PHO2* expression module components, the miR399 and *PHO2* transcripts, altered to differing degrees. However, in spite of the unique response of salt-stressed Col-0, *MIM399* and *MIR399* plants at the phenotypic, physiological and molecular levels, the expression of the PO_4_ transporter genes, *PHT1;4* and *PHT1;9*, was elevated in each plant line following the imposed stress. This finding suggests that miR399 abundance is altered in *Arabidopsis* in response to salt stress to catalyze downstream molecular alternations to the miR399/*PHO2* pathway to modify PO_4_ transporter activity to promote P translocation from the root to the shoot tissue of salt-stressed *Arabidopsis* plants. Altered PO_4_ transporter activity in salt stressed *Arabidopsis* plants would provide additional supplies of the essential cellular resource, P, which could be utilized to maintain essential biological functions and to mount an adaptive response to salt stress.

## 2. Results

### 2.1. Generation of Arabidopsis Transformant Lines Molecularly Modified to Have Altered miR399 Abundance

Considering the well documented regulatory role of miR399 in the adaptive response of *Arabidopsis* to growth in a P deficient environment [31,32,35,36,37,38], it was surprising to observe altered miR399 abundance in response to the abiotic stress challenges of salt and drought stress [19]. Therefore, to further investigate the role of miR399 in the response of *Arabidopsis* to salt stress, a molecular modification approach was initially used to generate transformant lines with altered miR399 levels. More specifically, an endogenous target mimicry (eTM) transgene [39,40] was introduced into wild-type *Arabidopsis* plants (ecotype Columbia-0 (Col-0)) via *Agrobacterium tumefaciens* (*Agrobacterium*)-mediated transformation to produce an *Arabidopsis* plant line, termed the *MIM399* plant, with reduced miR399 abundance. Similarly, *Agrobacterium*-mediated transformation was again utilized to introduce, and to overexpress, the miR399 precursor transcript (specifically the *PRE-MIR399C* precursor transcript) in Col-0 plants [31,32] to generate the *MIR399* transformant line; an *Arabidopsis* plant line constructed to have elevated miR399 abundance.

Previous research [41] has shown that *Arabidopsis* plants molecularly engineered to over-express the eTM specific for miR399, namely an artificially synthesized transgene version of the *Arabidopsis* non-protein-coding RNA, *INDUCED BY PHOSPHATE STARVATION1* (*IPS1*), did not display any obvious phenotypic variation to unmodified *Arabidopsis* plants during either the vegetative or reproductive phases of development. Comparison of Figure 1B to Figure 1A, and Figure 1E to Figure 1D, clearly shows that the constitutive expression of the miR399-specific eTM transgene in the selected *MIM399* transformant line confirmed this previous report [41] with *MIM399* plants phenotypically indistinguishable to Col-0 plants throughout the vegetative and reproductive phases of development. In pummelo (*Citrus grandis*) plants however, the expression of a miR399-specific eTM transgene resulted in the expression of readily observable phenotypic defects in the floral tissues, including smaller sized petals and sepals, reduced petal numbers, short anther filaments and reduced anther numbers [42]. However, the *Arabidopsis MIM399* plants generated in this study did not display any readily observable differences to Col-0 plants during their reproductive development. In contrast to the *MIM399* transformant line, the rosette of the *MIR399* plant line (Figure 1C) was slightly reduced in size compared to Col-0 rosettes (Figure 1A). Furthermore, in the distal tips of mature rosette leaves of the *MIR399* plant, areas of chlorotic and necrotic tissue formed (Figure 1C). In addition to this vegetative phenotype, the inflorescence stem and its siliques were a pale green, to yellow color (Figure 1F), compared to the uniform, healthy green coloration of these reproductive organs of Col-0 plants (Figure 1D). It is important to mention here however, that in spite of this phenotypic variation, the fertility of the *MIR399* plant was comparable to that of wild-type *Arabidopsis* plants. The development of chlorotic and necrotic tissues in the distal tips of rosette leaves of *Arabidopsis* plants molecularly modified to over-accumulate miR399, or in the *pho2* mutant background, an *Arabidopsis* plant line defective in the activity of the protein (PHO2) encoded by the miR399 target gene, *PHO2*, has been reported previously [31,32,35], and demonstrated to be the result of toxic P over-accumulation in the shoot tissues of these *Arabidopsis* lines when cultivated in a P replete environment.

To determine whether the *MIM399* and *MIR399* transformant lines selected for further experimental characterization had altered miR399 abundance, and/or *PHO2* target gene expression, RT-qPCR was used to quantify the levels of the miR399 and *PHO2* transcripts in *MIM399* and *MIR399* plants. This analysis revealed that miR399 abundance was reduced by 2.2-fold in the rosette leaves of *MIM399* plants, compared to its abundance in Col-0 rosette leaves (Figure 1G). In response to reduced miR399 abundance, *PHO2* target gene expression was also determined to be reduced to a similar degree, down by 2.4-fold, in the rosette leaves of *MIM399* plants. This formed an unexpected finding considering that the predominant mode of plant miRNA target gene expression regulation is messenger RNA (mRNA) cleavage [28,29]. Therefore, the level of the *PHO2* target transcript was expected to be increased in response to reduced miR399 abundance in *MIM399* plants, and not to have a reduced degree of expression as observed here (Figure 1G). This finding strongly indicated that translational repression also formed a mode of RNA silencing directed by miR399 to regulate *PHO2* expression in the *MIM399* plant line transformed with the eTM transgene, with target transcript abundance demonstrated to scale in accordance with level of the targeting miRNA when this alternate mechanism of RNA silencing is in operation in *Arabidopsis* [29,30]. Considering that the *MIR399* transformant line displayed the same vegetative phenotype as those previously reported for *Arabidopsis* plants molecularly modified to over-accumulate miR399, specifically the formation of areas of chlorosis and necrosis in mature rosette leaves [31,32,35], the 4.1-fold elevation to miR399 levels in *MIR399* plants was expected (Figure 1G). However, in response to this significant elevation to miR399 abundance in *MIR399* plants, the *PHO2* transcript level was determined to only be mildly reduced by 1.1-fold (Figure 1G). A very mild degree of reduction to the level of the *PHO2* target transcript in response to a significantly elevated level of miR399 in the *MIR399* transformant line again indicated that miRNA-directed target mRNA cleavage was not the only mode of RNA silencing directed by miR399 to control *PHO2* expression in the vegetative tissues of 25-day-old *Arabidopsis* plants molecularly modified to have altered miR399 abundance.

### 2.2. Phenotypic and Physiological Assessment of Control and Salt-Stressed Col-0, MIM399 and MIR399 Plants

In this study, Col-0, *MIM399* and *MIR399* seeds were germinated and cultivated under a standard growth regime on solid *Arabidopsis* growth medium for 8 days, at which time, equal numbers of 8-day-old seedlings were transferred to either (1) fresh plates of standard solid *Arabidopsis* growth medium, or (2) fresh plates that contained solid *Arabidopsis* growth medium that had been supplemented with 150 millimolar sodium chloride (150 mM NaCl). Post seedling transfer, the ‘*control*’ and ‘*salt-stress*’ plates were returned to the growth cabinet with seedlings cultivated for an additional 7-day period under a standard *Arabidopsis* growth regime. At day 15, control (termed Ns plants herein) and salt stressed (termed NaCl plants herein) Col-0, *MIM399* and *MIR399* seedlings were sampled for assessment of their; (i) fresh weight; (ii) rosette area; (iii) primary root length; (iv) anthocyanin abundance, and (v) chlorophyll *a* and *b* content. Measurement of this series of phenotypic and physiological metrics was conducted in an attempt to determine whether altered miR399 abundance in the *MIM399* and *MIR399* transformant lines provided either plant line with an altered degree of tolerance to the imposed stress.

Figure 2A shows that at 15 days of age, *MIM399*/Ns seedlings were slightly larger than Col-0/Ns seedlings, while *MIR399*/Ns seedlings were mildly reduced in size. However, apart from these mild differences in seedling size, the overall morphology of the rosette leaves of 15-day-old *MIM399*/Ns and *MIR399*/Ns seedlings remained unchanged from those of Col-0/Ns plants. The 7-day cultivation period on solid *Arabidopsis* growth medium supplemented with 150 mM NaCl was clearly detrimental to the developmental progression of Col-0, *MIM399* and *MIR399* plants (Figure 2A). More specifically, when compared to the rosettes of Col-0/Ns, *MIM399*/Ns and *MIR399*/Ns plants, the rosettes of 15-day-old Col-0/NaCl, *MIM399*/NaCl and *MIR399*/NaCl seedlings were reduced in their overall size due to a combination of the (1) development of smaller sized rosette leaves, (2) downwards curling of the distal tips of rosette leaves, and (3) inhibition of rosette leaf petiole elongation (Figure 2A). The 7-day salt stress treatment regime also resulted in the development of areas of chlorosis in the rosette leaves of Col-0/NaCl, *MIM399*/NaCl and *MIR399*/NaCl seedlings and clearly promoted the considerable over-accumulation of anthocyanin, with anthocyanin over-accumulation especially apparent in the tissues surrounding the shoot apex, including either emerging or newly emerged rosette leaves, and the petioles of fully emerged rosette leaves (Figure 2A).

When cultivated for 15 days on standard *Arabidopsis* growth medium, the average fresh weight of a Col-0 seedling was 28.1 milligrams (mg). In comparison, the fresh weight of a 15-day-old *MIM399*/Ns seedling was slightly heavier by 6.6% at 30.0 mg, whereas the fresh weight of a 15-day-old *MIR399*/Ns seedling was 24.1 mg to represent a 14.2% reduction in fresh weight compared to that of Col-0/Ns seedlings (Figure 2B). Considering the readily apparent negative impact that salt stress had on Col-0 development, it was unsurprising to subsequently determine that the fresh weight of Col-0/NaCl seedlings was reduced by 26.1% (20.8 mg), compared to the fresh weight of Col-0/Ns seedlings (Figure 2A,B). Interestingly, both of the generated transformant lines showed larger reductions to their fresh weight following their exposure to salt stress. Namely, the fresh weight of *MIM399*/NaCl and *MIR399*/NaCl seedlings was reduced by 35.9 (19.2 mg) and 35.6% (15.5 mg), compared to the fresh weights of their control grown counterparts, *MIM399*/Ns and *MIR399*/Ns plants, respectively (Figure 2B). This result suggested that any alteration to the level of miR399 from its wild-type abundance, enhanced the sensitivity of *Arabidopsis* to salt stress.

To further investigate the negative impact of salt stress on Col-0, *MIM399* and *MIR399* plant development, we next determined the average rosette area of 15-day-old Col-0/Ns, *MIM399*/Ns, *MIR399*/Ns, Col-0/NaCl, *MIM399*/NaCl and *MIR399*/NaCl seedlings (Figure 2C). The average rosette area of a Col-0/Ns plant was 18.2 mm^2^. As readily apparent in Figure 2A, it was unsurprising to determine that the rosette area of *MIM399*/Ns plants was increased by 12.6% to 20.5 mm^2^ (Figure 2C). Additionally, based on the phenotypes presented in Figure 2A, the rosette area of *MIR399*/Ns seedlings was revealed to be reduced by 21.4% to 14.3 mm^2^ (Figure 2C). The 7-day cultivation period of 8-day-old Col-0 seedlings in the presence of 150 mM NaCl was subsequently determined to reduce the rosette area of Col-0/NaCl plants by 46.6% to 9.7 mm^2^. The degree of impairment of rosette development was revealed to be similar for *MIM399*/NaCl plants, reduced by 45.5% to 11.2 mm^2^ (Figure 2C). In contrast to salt-stressed Col-0 and *MIM399* plants, the rosette area of *MIR399*/NaCl plants was only moderately reduced by 27.4% to 10.4 mm^2^. This milder degree of impact on rosette development could be the result of (1) the *MIR399* transformant line engineered to have elevated miR399 abundance being less sensitive to the applied stress, or (2) the already impaired vegetative development of *MIR399*/Ns plants masked the degree of impact that the applied stress had on *MIR399*/NaCl rosette development.

Altered root architecture is a widely documented consequence of the exposure of *Arabidopsis* to salt stress [43,44]. Therefore, the primary root length of Col-0/NaCl, *MIM399*/NaCl and *MIR399*/NaCl seedlings was compared to that of their control grown counterparts. The primary root length of Col-0/NaCl plants was determined to be 22.2 mm, reduced by 54.4% compared to the primary root length of 48.6 mm of Col-0/Ns seedlings (Figure 2D). In comparison, the primary root length of *MIM399*/NaCl plants (33.7 mm) was only reduced by 31.0% compared to the primary root length of *MIM399*/Ns plants (48.8 mm). The degree of reduction to primary root elongation in *MIR399*/NaCl plants was determined to sit in between that of Col-0/NaCl and *MIM399*/NaCl plants (Figure 2D). More specifically, the primary root length of *MIR399*/NaCl plants was reduced by 39.3% to 25.9 mm, compared to 42.6 mm for *MIR399*/Ns plants.

Post the assessment of the three phenotypic metrics of fresh weight, rosette area and primary root length, the physiological parameters of anthocyanin abundance (Figure 2E) and chlorophyll *a* (Figure 2F) and *b* (Figure 2G) content were analyzed next. The abundance of anthocyanin in Col-0/Ns, *MIM399*/Ns and *MIR399*/Ns seedlings was highly similar (Figure 2E). More specifically, the anthocyanin abundance of Col-0/Ns seedlings was determined to be 2.2 micrograms per gram of fresh weight (2.2 μg/g FW). Similarly, the anthocyanin abundance of *MIM399*/Ns and *MIR399*/Ns plants was revealed to be slightly reduced to approximately 2.1 μg/g FW (Figure 2E). A similar degree of elevation in the abundance of anthocyanin, namely an 80.8 and 76.8% elevation, was observed in Col-0/NaCl and *MIM399*/NaCl plants respectively. In contrast, salt stress induced a 163.2% increase to the abundance of anthocyanin in *MIR399*/NaCl seedlings (Figure 2E); an observation that is clearly visible in Figure 2A, and which strongly indicated that the anthocyanin biosynthesis pathway is promoted to a much higher degree in response to the imposed stress in the *MIR399* transformant line, compared to the level of promotion of this antioxidant pigment production pathway in either Col-0 plants or the *MIM399* transformant line.

Inhibition of the photosynthetic capacity of a plant is a reported consequence of salt stress [45]. Therefore, the chlorophyll *a* and *b* content of salt stressed Col-0, *MIM399* and *MIR399* plants was next assessed for comparison to the control grown counterpart of each plant line. The chlorophyll *a* content of Col-0/Ns seedlings was determined to be 0.73 milligrams per gram of fresh weight (0.73 mg/g FW). A mildly elevated chlorophyll *a* content, an 8.5 and 12.4% increase respectively, was documented for *MIM399*/Ns and *MIR399*/Ns seedlings (Figure 2F). The 7-day 150 mM NaCl stress regime resulted in the chlorophyll *a* content of Col-0/NaCl seedlings being reduced by 20.8% to 0.59 mg/g FW. A slightly higher degree of reduction, 29.4%, to chlorophyll *a* content was observed for *MIM399*/NaCl seedlings. The extent to which chlorophyll *a* was reduced in *MIR399*/NaCl plants, however, was more than double that determined for Col-0/NaCl plants with chlorophyll *a* content decreased by 42.0% (Figure 2F).

The abundance profile constructed for chlorophyll *b* across the control and salt-stressed *Arabidopsis* lines was highly similar to the chlorophyll *a* profile documented for the three assessed plant lines. Specifically, the chlorophyll *b* content of *MIM399*/Ns and *MIR399*/Ns plants was mildly elevated by 13.8 and 16.0% respectively, compared to the chlorophyll *b* content of 0.18 mg/g FW for Col-0/Ns seedlings (Figure 2G). Furthermore, the 7-day cultivation period in the presence of 150 mM NaCl reduced the chlorophyll *b* content of Col-0, *MIM399* and *MIR399* plants to differing degrees, namely, chlorophyll *b* content was reduced by 40.0, 48.5 and 57.9% in Col-0/NaCl, *MIM399*/NaCl and *MIR399*/NaCl plants, respectively (Figure 2G). Taken together, the physiological assessments performed on Col-0, *MIM399* and *MIR399* plants, including the determination of anthocyanin (Figure 2E), chlorophyll *a* (Figure 2F) and *b* (Figure 2G) content, suggested that at the physiological level, the *MIR399* transformant line was the most sensitive of the three *Arabidopsis* lines assessed in this study to the imposed 7-day salt stress treatment regime.

### 2.3. Quantification of miR399 Abundance and the Level of PHO2 Target Gene Expression in Control and Salt-Stressed Col-0, MIM399 and MIR399 Plants

Considering that the phenotypic and physiological analyses presented in Figure 2 clearly show that Col-0, *MIM399* and *MIR399* plants responded differently to the 7-day cultivation period in the presence of 150 mM NaCl, we next sort to characterize the response of each plant line to salt stress at the molecular level. In our previous study where we demonstrated that miR399 is responsive to salt stress [19], we used the well documented stress-induced gene, *Δ1-PYRROLINE-5-CARBOXYLATE SYNTHETASE1* (*P5CS1*; *AT2G39800*) [46] to show that exposed plants were in fact experiencing a degree of stress at the molecular level. Therefore, RT-qPCR was initially used to quantify *P5CS1* expression in control and salt stressed Col-0, *MIM399* and *MIR399* plants (Figure 3A). Compared to Col-0/Ns plants, *P5CS1* expression remained unchanged in *MIM399*/Ns plants and was moderately reduced by 1.5-fold in *MIR399*/Ns seedlings. The 7-day cultivation period in the presence of 150 mM NaCl was next revealed by RT-qPCR to elevate the level of *P5CS1* expression by 2.6-fold in Col-0/NaCl plants, compared to Col-0/Ns seedlings (Figure 3A). Similarly, in *MIM399*/NaCl and *MIR399*/NaCl plants, *P5CS1* expression was determined to be increased by 3.1- and 2.3-fold, compared to *MIM399*/Ns and *MIR399*/Ns seedlings, respectively. Taken together, the induction of *P5CS1* expression in salt-stressed Col-0, *MIM399* and *MIR399* plants (Figure 3A), clearly showed that the three analyzed *Arabidopsis* lines were experiencing stress at the molecular level resulting from their cultivation for a 7-day period in the presence of 150 mM NaCl.

RT-qPCR was next used to (1) confirm that miR399 is responsive to salt stress, and (2) determine if the degree of miR399 responsiveness to the imposed stress differed in either of the transformant lines molecularly modified to have altered miR399 abundance. RT-qPCR revealed that in 15-day-old control *MIM399* and *MIR399* seedlings, miR399 abundance was reduced by 1.9-fold and elevated by 3.9-fold respectively (Figure 3B). It is important to note here that miR399 abundance was demonstrated to be reduced and elevated to greater degrees in the rosette leaves of 25-day-old soil grown *MIM399* and *MIR399* plants (Figure 1G). The RT-qPCR analysis reported in Figure 3B was however conducted on 15-day-old *MIM399*/Ns and *MIR399*/Ns whole seedlings which had been cultivated on standard *Arabidopsis* growth media. Therefore, the contrasting results presented in Figure 1G and Figure 3B is likely the combined result of the difference in the (1) plant age (2) tissue sampled, and (3) growth conditions, prior to RT-qPCR quantification of miR399 abundance being performed. Compared to Col-0/Ns seedlings, miR399 abundance was elevated by 2.4-fold in Col-0/NaCl seedlings (Figure 3B). This finding confirmed our previous demonstration that in 15-day-old Col-0 whole seedlings, the abundance of the miR399 sRNA was elevated by the cultivation of 8-day-old Col-0 seedlings for a 7-day period in the presence of 150 mM NaCl [19]. Post obtainment of this confirmation, RT-qPCR was next used to profile miR399 abundance in *MIM399*/NaCl and *MIR399*/NaCl plants. In *MIM399*/NaCl plants, miR399 abundance was significantly elevated by 3.6-fold compared to its level in *MIM399*/Ns seedlings (Figure 3B). RT-qPCR next showed that compared to the significant enhancement to the miR399 level in Col-0/NaCl and *MIM399*/NaCl plants, miR399 abundance was only moderately elevated by 1.6-fold in *MIR399*/NaCl plants (Figure 3B). However, it is important to note here that the level of miR399 was already elevated in the *MIR399* transformant line, and therefore, only moderate enhancement to the abundance of this sRNA in salt-stressed *MIR399* transformants, compared to its significant degree of elevation in salt-stressed Col-0 and *MIM399* plants, was expected. Nonetheless, elevated miR399 abundance in 15-day-old Col-0, *MIM399* and *MIR399* seedlings post their exposure to the 7-day cultivation period in the presence of 150 mM NaCl confirmed miR399 to be a salt stress responsive miRNA.

Post the demonstration that miR399 abundance is elevated in response to salt stress, we next quantified the expression level of the miR399 target gene, *PHO2*, by RT-qPCR. In response to the 1.9-fold reduction to miR399 levels in *MIM399*/Ns plants, it was curious to observe that the abundance of the *PHO2* target transcript was decreased by a similar degree, down by 2.0-fold (Figure 3C). This formed an unexpected finding considering that we have previously shown that in Col-0 plants, miR399 regulates the expression of its *PHO2* target gene via the canonical mechanism of plant miRNA target gene expression regulation, target mRNA cleavage [19,47]. Considering this finding, decreased miR399 abundance in control grown *MIM399* plants was expected to result in the release of miR399-directed *PHO2* cleavage, and therefore, elevated *PHO2* transcript abundance. However, reduced *PHO2* expression as reported in Figure 3C suggests that miRNA-directed target mRNA cleavage is not the predominant mode of RNA silencing directed by miR399 to regulate *PHO2* expression in the *MIM399* transformant line. Adding further weight to this suggestion is the finding that the 3.9-fold elevation in miR399 abundance in *MIR399*/Ns plants, only resulted in a very mild 1.1-fold reduction to the level of *PHO2* expression. Therefore, again, if target mRNA cleavage was the predominant mode of regulation directed by miR399 to control *PHO2* expression in *MIR399* plants, then significantly reduced *PHO2* transcript abundance would have been expected to have been detected in the *MIR399*/Ns sample. When taken together, the miR399 abundance and *PHO2* expression trends documented for *MIM399*/Ns and *MIR399*/Ns plants (Figure 3C) strongly suggested that translational repression is the predominant mode of target gene expression regulation directed by miR399 to control *PHO2* transcript abundance in these two *Arabidopsis* transformant lines which had been molecularly modified to have altered miR399 levels.

In Col-0/NaCl plants, and when compared to Col-0/Ns plants, *PHO2* expression was reduced by 4.2-fold (Figure 3C). Reduced *PHO2* expression was expected in Col-0/NaCl seedlings considering that miR399 abundance was elevated by 2.4-fold (Figure 3B). This observation also confirmed our previous findings [19,47] that the abundance of the *PHO2* target transcript is regulated by miR399 via the mRNA cleavage mode of RNA silencing in 15-day-old wild-type *Arabidopsis* seedlings following their cultivation for a 7-day period in the presence of 150 mM NaCl. As observed for *MIM399*/Ns and *MIR399*/Ns plants, an atypical *PHO2* expression profile was constructed for *MIM399*/NaCl seedlings. More specifically, *PHO2* transcript abundance was mildly elevated by 1.2-fold (Figure 3C) in response to the significant 3.6-fold enhancement to the abundance of the targeting sRNA, miR399 (Figure 3B). This unexpected transcript profile again suggested that in the *MIM399* transformant line which had been molecularly modified to have reduced miR399 abundance, the mechanism of RNA silencing directed by miR399 to regulate *PHO2* expression transitioned from the canonical form of plant miRNA-directed target gene expression regulation, mRNA cleavage (as observed in Col-0 seedlings), to the alternate form of miRNA-directed expression regulation in plants, translational repression. Alternatively, the eTM transgene (that is; the artificially engineered *IPS1* sequence) specific to miR399 introduced into the *MIM399* plant line, caused the entire miR399/*PHO2* expression module to become dysfunctional. In response to the mild degree (1.6-fold) of enhanced miR399 abundance in *MIR399*/NaCl plants, compared to *MIR399*/Ns plants (Figure 3B), *PHO2* expression was revealed by RT-qPCR to be significantly reduced by 4.0-fold (Figure 3C). It is important to note here that in *MIR399*/Ns plants, when compared to Col-0/Ns plants, the significant 3.9-fold elevation to miR399 abundance (Figure 3B), only resulted in a mild 1.1-fold reduction to the level of the *PHO2* transcript (Figure 3C). Therefore, the significant reduction to *PHO2* transcript abundance in response to the mild additional elevation to miR399 levels in *MIR399*/NaCl plants (compared to *MIR399*/Ns seedlings), indicated that a threshold may have existed in this transformant line that needed to be surpassed in order for miR399 to exert a regulatory effect over the expression of its *PHO2* target transcript via a mRNA cleavage mode of RNA silencing. However, the sRNA/target gene relationship for the miR399/*PHO2* expression profile in the *MIR399* transformant line still differed from that constructed for unmodified Col-0 plants to further suggest that the introduction of the miR399 overexpression transgene into wild-type *Arabidopsis* via *Agrobacterium*-mediated transformation caused some degree of dysfunction of the miR399/*PHO2* expression module.

### 2.4. RT-qPCR Profiling of the Response to Salt Stress of Upstream and Downstream Elements of the miR399/PHO2 Pathway

The activity of the transcription factor, PHOSPHATE RESPONSIVE1 (PHR1), has been demonstrated to be promoted by the cultivation of *Arabidopsis* plants in a PO_4_ deplete environment [36] in order to regulate the expression of PO_4_ responsive genes as part of the attempt of the *Arabidopsis* plant to maintain P homeostasis. Therefore, RT-qPCR was utilized to assess *PHR1* gene expression in 15-day-old Col-0, *MIM399* and *MIR399* seedlings following their cultivation for a 7-day period in the presence of 150 mM NaCl to determine at what point in the entire miR399/PHO2 pathway the salt stress was imposing its effect. This analysis revealed that compared to Col-0/Ns plants, the expression of the *PHR1* transcription factor was mildly reduced by 1.1- and 1.2-fold in *MIM399*/Ns and *MIR399*/Ns seedlings, respectively (Figure 4A). In Col-0/NaCl plants, *PHR1* expression was reduced by 1.9-fold, compared to its expression level in Col-0/Ns seedlings. Compared to *MIM399*/Ns and *MIR399*/Ns plants, *PHR1* expression was mildly elevated by 1.2-fold in *MIM399*/NaCl seedlings and mildly decreased by 1.3-fold in *MIR399*/NaCl seedlings, respectively (Figure 4A). The milder degree of alteration to the *PHR1* expression profile constructed for *MIM399*/NaCl and *MIR399*/NaCl seedlings, compared to that obtained for Col-0/NaCl plants, indicated that the transcriptional activity of the *PHR1* locus is modulated differently, and to different degrees, by the imposed stress at the start of the miR399/*PHO2* pathway.

Having demonstrated that the abundance of the *PHR1* transcript was mildly altered in *MIM399*/Ns, *MIR399*/Ns, *MIM399*/NaCl and *MIR399*/NaCl plants, and moderately reduced by 1.9-fold in Col-0/NaCl plants, we next sought to determine if there were expression changes to the downstream targets of PHO2-directed ubiquitination [35,36,37], namely; assessment of the transcriptional activity of the PO_4_ transporter encoding genes, *PHT1;4* and *PHT1;9*. In *MIM399*/Ns and *MIR399*/Ns seedlings, *PHT1;4* expression was revealed to be significantly elevated by 4.2- and 5.3-fold, respectively (Figure 4B). In Col-0/NaCl seedlings, *PHT1;4* expression was also significantly enhanced by 4.6-fold. In *MIM399*/NaCl plants, compared to *MIM399*/Ns plants, *PHT1;4* transcript abundance was highly elevated by 10.7-fold (Figure 4B). RT-qPCR additionally revealed *PHT1;4* expression to be elevated by 6.0-fold in *MIR399*/NaCl plants, compared to its expression level in *MIR399*/Ns plants (Figure 4B). A similar expression profile to that obtained for *PHT1;4* was constructed for *PHT1;9* in *MIM399*/Ns and *MIR399*/Ns seedlings (Figure 4C). Namely, *PHT1;9* expression was significantly elevated by 3.8- and 6.1-fold in *MIM399*/Ns and *MIR399*/Ns seedlings, respectively (Figure 4C). In contrast to elevated *PHT1;9* expression in *MIM399*/Ns and *MIR399*/Ns plants, *PHT1;9* transcript abundance was moderately decreased by 1.7-fold in Col-0/NaCl seedlings. In *MIM399*/NaCl plants, compared to *MIR399*/Ns plants, *PHT1;9* expression was further significantly elevated by 3.5-fold (Figure 4C). In contrast to the *MIM399* transformant line, yet in agreement with Col-0/NaCl seedlings, *PHT1;9* transcript abundance was mildly reduced by 1.4-fold in *MIR399*/NaCl seedlings, compared to its expression level in *MIR399*/Ns seedlings (Figure 4C). Taken together, the results presented in Figure 4, in addition to the data presented in Figure 3B,C, strongly suggested that the entire miR399/*PHO2* pathway, and not just the miR399/*PHO2* expression module central to this pathway, is responsive to salt stress.

## 3. Discussion

Stemming from our previous demonstration that miR399 abundance is altered in response to salt and drought stress [19], in addition to the well documented requirement of altered miR399 abundance for *Arabidopsis* to attempt to maintain P homeostasis when cultivated in a PO_4_ deplete environment [19,31,32,35,36,37,48], we adopted two transgene-based technologies [31,32,40,41,42] to generate *Arabidopsis* transformant lines with altered miR399 abundance. Post the generation of *MIM399* and *MIR399* plants, these two transformant lines, along with wild-type *Arabidopsis* (Col-0) plants, were exposed to a 7-day cultivation period in the presence of 150 mM NaCl at 8 days of age to determine if either transformant line displayed differences in its phenotypic, physiological or molecular response to that of unmodified Col-0 plants to the imposed stress. This approach was adopted in this study to attempt to further identify the requirement of altered miR399 abundance for *Arabidopsis* to mount an adaptive response to salt stress.

### 3.1. The Generation of Arabidopsis Transformant Lines Molecularly Modified to Have Altered miR399 Abundance

Figure 1 clearly shows that in both the vegetative and reproduction phases of *Arabidopsis* development, *MIM399* plants were largely indistinguishable to unmodified Col-0 plants. Although it has been reported previously that the constitutive expression of the eTM specific to miR399 in *Arabidopsis* failed to result in the generated transformant lines displaying any phenotypic differences to wild-type *Arabidopsis* plants [39,40], we subsequently employed RT-qPCR to confirm that the eTM transgene introduced into the *MIM399* transformant line did indeed result in the molecular alteration of the miR399/*PHO2* expression module. This analysis confirmed that the abundance of miR399 was reduced by 2.2-fold in the rosette leaves of 25-day-old *MIM399* plants. RT-qPCR further revealed that in response to reduced miR399 abundance in *MIM399* rosette leaves, the expression of *PHO2* was reduced to a similar degree, down by 2.4-fold (Figure 1G). Elevated *PHO2* gene expression in response to a reduced miR399 level was expected in the *MIM399* transformant line considering that we have previously demonstrated the predominant mode of miR399-directed regulation of *PHO2* expression in *Arabidopsis* vegetative tissues to be the target mRNA cleavage mechanism of RNA silencing [19,47]. However, the observed reduction to both the level of miR399 and of its target transcript, *PHO2*, in the rosette leaves of 25-day-old control grown *MIM399* plants suggested that the introduced molecular alteration to the miR399/*PHO2* expression module in *MIM399* plants had resulted in the transition from a mRNA cleavage, to a translational repression mode of RNA silencing being directed by miR399 to regulate the abundance of the *PHO2* transcript in *MIM399* plants; a mode of expression regulation previously suggested [35] to be directed by miR399 to control *PHO2* transcript levels in *Arabidopsis*. Alternatively, *PHO2* gene expression in *MIM399* plants may have been modulated by a gene expression regulatory mechanism independent of miR399-directed expression regulation as a result of the molecular modifications made to the miR399/*PHO2* expression module in the *MIM399* transformant line.

In direct contrast to the *MIM399* transformant line, the *MIR399* plant line which was molecularly engineered to overexpress the *PRE-MIR399C* precursor sequence, and therefore, to have elevated levels of miR399 abundance, readily displayed phenotypic differences to unmodified Col-0 plants during both the vegetative and reproductive phases of development. Specifically, the rosette of 25-day-old *MIR399* plants was slightly reduced in size compared to the rosette of Col-0 plants of the same age (Figure 1A,C); a reduction in rosette area that stemmed from the development of rosette leaves that were broader due to their obovate shape, compared to the thinner and elongated lanceolate shape of Col-0 rosette leaves (Figure 1A,C). In addition, areas of chlorotic and necrotic tissue formed at the distal tips of mature rosette leaves of *MIR399* plants (Figure 1C). Furthermore, the inflorescence stem and the siliques of the *MIR399* plant displayed a pale green, to dull yellow coloration (Figure 1F), compared to the uniform healthy bright green color of the primary inflorescence and siliques of Col-0 plants (Figure 1D). These phenotypic consequences have been reported previously [31,32,35] in *Arabidopsis* plant lines molecularly modified to have elevated miR399 abundance, or that harbor a knockout insertion mutation in the *PHO2* locus (the *pho2* mutant line); phenotypes demonstrated to be the result of the toxic over-accumulation of P in the shoot tissues of these plant lines when cultivated in a standard growth environment where P is not in limited supply [31,32,35]. Although the *MIR399* transformant displayed these previously reported developmental consequences of overexpressing miR399, RT-qPCR was still applied to profile any molecular alteration of the miR399/*PHO2* expression module. Compared to its abundance in the rosette leaves of 25-day-old Col-0 plants, the level of miR399 was revealed to be elevated by 4.1-fold in *MIR399* rosette leaves. RT-qPCR further revealed, that in spite of the significant elevation in miR399 abundance in *MIR399* rosette leaves, the expression of its *PHO2* target gene was only mildly repressed by 1.1-fold (Figure 1G). This finding adds further weight to the suggestion that molecular modifications targeting the miR399/*PHO2* expression module via a transgene-based approach in *Arabidopsis*, mediate the transition of the mode of RNA silencing directed by miR399 to control *PHO2* target transcript abundance from the standard mRNA cleavage mode, to a translational repression mode; a mode of RNA silencing previously suggested to be directed in *Arabidopsis* by miR399 to regulate the abundance of its *PHO2* target transcript [35].

### 3.2. Profiling of the Phenotypic, Physiological and Molecular Response of Col-0, MIM399 and MIR399 Plants Exposed to Salt Stress

Visual inspection revealed that at 15 days of age, *MIM399*/Ns seedlings were larger than Col-0/Ns seedlings (Figure 2A). This observation was confirmed via quantification of the fresh weight and rosette area of Col-0/Ns and *MIM399*/Ns seedlings (Figure 2B,C). RT-qPCR subsequently revealed that *P5CS1* expression remained at wild-type equivalent levels in *MIM399*/Ns seedlings (Figure 3A); a finding that suggested that the promotion of *MIM399*/Ns rosette development was not the result of this transformant line responding to a perceived ”*stress*” signal via the introduction and constitutive expression of the miR399-specific eTM transgene. However, RT-qPCR did show that the abundance of both the miR399 sRNA, and its *PHO2* target transcript were reduced by a similar degree (-1.9- and -2.0-fold, respectively) in *MIM399*/Ns seedlings (Figure 3B,C). RT-qPCR next revealed that in response to decreased *PHO2* expression, and therefore PHO2 protein abundance, the expression of the two PO_4_ transporters, *PHT1;4* and *PHT1;9*, was elevated in *MIM399*/Ns seedlings (Figure 4B,C). The abundance of the PHT1;4 and PHT1;9 PO_4_ transporter proteins is controlled in *Arabidopsis* via PHO2-mediated ubiquitination [31,32,35,36,37]. Therefore, elevated *PHT1;4* and *PHT1;9* expression in *MIM399*/Ns seedlings potentially stemmed from the relaxation of the degree of ubiquitination of these two PO_4_ transporter proteins in *MIM399*/Ns plants where *PHO2* gene expression was reduced (Figure 3C). This in turn, would be expected to lead to promotion of Pi root-to-shoot translocation, and thus, an increased abundance of Pi in the aerial tissues of *MIM399*/Ns seedlings [48,49,50]. For cellular processes such a photosynthesis, Pi forms one of the most central molecular building blocks, therefore, elevated Pi in the aerial tissues of *MIM399*/Ns seedlings could potentially account for the increased size of the rosette of this transformant line, compared to Col-0 seedlings of the same age (Figure 2A,C). Adding to the suggestion that enhanced Pi root-to-shoot translocation in *MIM399*/Ns seedlings accounted for the increased size of their rosettes, was our finding that the primary root length of *MIM399*/Ns seedlings remained unchanged compared to the length of the primary root of Col-0/Ns plants (Figure 2D). Further support of this theory is additionally provided by the findings of [51], which showed that mild doses of Pi (1.25 to 10.0 μM) promoted the development of the rosette of wild-type *Arabidopsis* plants.

At 15 days of age, and only when cultivated under a standard growth regime, the *MIR399* transformant line displayed an opposing phenotype to that reported for *MIM399*/Ns seedlings (Figure 1A). More specifically, compared to Col-0/Ns seedlings, *MIR399*/Ns seedlings were reduced in size as evidenced by the documented decreases to the fresh weight (Figure 1B), rosette area (Figure 1C), and primary root length (Figure 1D) phenotypic metrics. Taken together, the mild impairment to *MIR399*/Ns seedling development, indicated that this transformant line was experiencing a degree of stress, even when cultivated under a standard growth regime. Therefore, RT-qPCR was employed to quantify the expression of the *P5CS1* stress marker gene in *MIR399*/Ns plants. Compared to Col-0/Ns seedlings, RT-qPCR revealed altered *P5CS1* expression in the *MIR399*/Ns sample (Figure 3A), a finding which suggested that the molecular modifications introduced into this transformant line resulted in the *MIR399* seedlings perceiving a mild degree of stress even when cultivated in a standard growth environment. RT-qPCR next revealed that miR399 abundance was elevated by 3.9-fold in *MIR399*/Ns seedlings (Figure 3B). However, *PHO2* target gene expression remained largely unresponsive to this significant elevation to miR399 abundance (Figure 3C). This finding indicated that in this transformant line generated to constitutively overexpress the *PRE-MIR399C* precursor transcript, the mode of RNA silencing directed by miR399 to regulate *PHO2* transcript abundance, transitioned from the canonical mRNA cleavage mode, to a translational repression mode. Translational repression as the predominant mode of *PHO2* expression regulation directed by miR399 in the *MIR399* transformant line was further supported by the significant 5.3- and 6.1-fold upregulated expression of the PO_4_ transporter genes *PHT1;4* and *PHT1;9*, respectively (Figure 4B,C). The observed degree of elevation to the abundance of the *PHT1;4* and *PHT1;9* transcripts strongly suggested that the PHO2-directed, ubiquitination-mediated regulation of PHT1;4 and PHT1;9 protein activity [31,32,35,36,37,49] had become completely defective in the *MIR399* transformant line. Dysfunctional regulation of PHT1;4 and PHT1;9 protein activity at the posttranslational level in *MIR399*/Ns seedlings, would in turn result in dysregulation of root-to-root Pi translocation and the toxic overaccumulation of Pi in the aerial tissues of this transformant line. This hypothesis is strongly supported by the reduced size of 15-day-old *MIR399*/Ns seedlings (Figure 2), and the subsequent development of regions of chlorotic and necrotic tissues in the distal tips of mature rosette leaves of 25-day-old soil grown *MIR399* plants (Figure 1); phenotypic consequences reported previously in *Arabidopsis* plant lines harboring molecular modification of the miR399/*PHO2* expression module [31,32,35].

We have previously demonstrated that miR399 abundance is altered by the cultivation of 8-day-old Col-0 seedlings for a 7-day period in the presence of 150 mM NaCl [19]. Here we confirm this result via RT-qPCR which revealed miR399 abundance to be elevated by 2.4-, 3.6- and 1.6-fold in Col-0/NaCl, *MIM399*/NaCl and *MIR399*/NaCl seedlings, respectively (Figure 3B). Elevated miR399 abundance in the three assessed *Arabidopsis* lines following their exposure to salt stress confirmed that this PO_4_ stress responsive miRNA [31,32,33,34,35,36,37], is also responsive to salt stress. RT-qPCR subsequently showed that in response to elevated miR399 abundance in Col-0/NaCl seedlings, *PHO2* gene expression was repressed by 4.2-fold (Figure 3C). This finding further confirms our previous report [19] that in salt-stressed Col-0 seedlings, miR399 controls the level its *PHO2* target transcript via a mRNA cleavage mode of RNA silencing. Further profiling of the response of upstream and downstream elements of the miR399/*PHO2* pathway in Col-0/NaCl seedlings revealed that *PHR1*, *PHT1;4* and *PHT1;9* expression was altered by -1.9-, 4.6- and -1.7-fold, respectively (Figure 4). Altered expression of miR399/*PHO2* pathway elements upstream and downstream of the central miR399/*PHO2* expression module itself, strongly suggested that all pathway elements are responsive to salt stress. Such a molecular alteration potentially ensures that an adequate supply of Pi continues to be translocated from the roots to the shoots of *Arabidopsis* plants via the xylem when *Arabidopsis* is exposed to salt stress in an attempt by the *Arabidopsis* plant to continue to maintain the function of essential biological processes while simultaneously mounting an adaptive response to the imposed stress.

In response to the 3.6-fold elevation to miR399 abundance in *MIM399*/NaCl seedlings (Figure 3B), *PHO2* expression was only mildly elevated by 1.2-fold (Figure 3C). This atypical miRNA/target gene expression profile again suggested that either (1) the mechanism of RNA silencing directed by miR399 to control *PHO2* transcript abundance in this transformant line had transitioned from the canonical mRNA cleavage mode of RNA silencing documented in Col-0/NaCl plants, to a translational repression mode of silencing, or (2) the constitutive overexpression the eTM transgene in the *MIM399* transformant line resulted in the complete dysfunction of the miR399/*PHO2* expression module. RT-qPCR analysis next revealed that in *MIM399*/NaCl seedlings, the expression of the upstream pathway element, *PHR1*, and of the downstream pathway elements, *PHT1;4* and *PHT1;9*, was elevated by 1.2-, 10.7- and 3.5-fold, respectively (Figure 4). Interestingly, of the three *Arabidopsis* plant lines exposed to salt stress, *PHR1* gene expression was only promoted in the *MIM399* transformant line. Furthermore, elevated *PHR1* gene expression appeared to positively influence *PHT1;9* expression, with elevated *PHT1;9* transcript abundance only detected in the *MIM399*/NaCl sample. However, elevated *PHT1;9* transcript abundance, and therefore PHT1;9 protein level, again suggests that expression alterations occur at each stage in the miR399/*PHO2* pathway to promote the translocation of Pi to the aerial tissues of salt-stressed *Arabidopsis* plants to be used as an essential resource, both to ensure the continuation of normal biological processes, and as part of the adaptive response of *Arabidopsis* to salt stress.

As reported for Col-0/NaCl seedlings, elevated (1.6-fold) miR399 abundance in *MIR399*/NaCl seedlings resulted in *PHO2* expression being decreased by 1.3-fold (Figure 3B,C); an opposing expression trend which indicated that *PHO2* transcript abundance is controlled via a mRNA cleavage mode of miR399-directed RNA silencing in the *MIR399* transformant line upon its exposure to salt stress. RT-qPCR profiling of *PHR1*, *PHT1;4* and *PHT1;9* transcript abundance showed that compared to *MIR399*/Ns plants, *PHR1* and *PHT1;9* expression was mildly decreased by 1.3- and 1.4-fold respectively (Figure 4A,C), whereas the expression of *PHT1;4* was significantly elevated by 6.0-fold (Figure 4B). This result again suggested that the altered expression of upstream and downstream elements of the miR399/*PHO2* pathway is required in *Arabidopsis* to alter the rate of Pi translocation from the root to shoot tissues. In the case of salt-stressed *MIR399* seedlings, enhancement to PHT1;4-mediated Pi translocation to the aerial tissues of this transformant line would promote the ability of the *MIR399* plant line to mount an adaptive response to the imposed stress, while simultaneously enabling the provision of additional cellular resources to ensure that other essential biological processes, such as photosynthesis, can be maintained during the stress exposure period.

### 3.3. Molecular Modification of the miR399/PHO2 Expression Module Alters the Response of Arabidopsis to Salt Stress

The molecular modifications to the miR399/*PHO2* expression module introduced into the Col-0 background resulted in both the *MIM399* and *MIR399* transformant lines displaying greater degrees of impact to aerial tissue development than did unmodified Col-0 seedlings by the salt stress treatment regime (Figure 2B–G). More specifically, the fresh weight of 15-day-old *MIM399*/NaCl and *MIR399*/NaCl seedlings was reduced by the same degree, an approximate 36% reduction (Figure 2B). In comparison, the fresh weight of Col-0 seedlings was only reduced by 26.1% by the imposed stress. In addition, the rosette area of *MIM399*/NaCl seedlings was reduced by 45.5%, a much higher degree of reduction than observed for either Col-0/NaCl (26.6%) or *MIR399*/NaCl (27.4%) seedlings (Figure 2C). Taken together, these two phenotypic measurements tentatively revealed that the shoot of the *MIM399* transformant line was most sensitive to the imposed stress. An inverse phenotypic response was however observed for the primary root, that is; the primary root length of Col-0/NaCl seedlings was significantly reduced by 54.4%, whereas the primary root length of *MIM399*/NaCl and *MIR399*/NaCl seedlings was, by comparison, only mildly reduced by 31.0% and 39.0%, respectively (Figure 2D). This finding suggested that the molecular modifications made to the miR399/*PHO2* expression module in *MIM399* and *MIR399* plants provided the root system of these two transformant lines with a mild degree of tolerance to the imposed stress. At the physiological level, the quantification of the abundance of anthocyanin (Figure 2E), chlorophyll *a* (Figure 2F) and chlorophyll *b* (Figure 2G) readily identified the *MIR399* transformant line to be the most sensitive to the imposed 7-day salt stress treatment regime. The enhanced physiological sensitivity of the *MIR399* transformant line to salt stress is most readily evidenced by the greater than 2-fold promotion to anthocyanin abundance in the rosette leaves of *MIR399*/NaCl seedlings, versus the comparatively mild degrees of elevation to the abundance of this pigment observed in Col-0/NaCl and *MIR399*/NaCl rosette leaves.

Although the data presented in Figure 2 clearly revealed that each of the three *Arabidopsis* plant lines assessed in this study uniquely responded at both the phenotypic and physiological level to cultivation for a 7-day period in the presence of 150 mM NaCl, in addition to displaying plant line specific miR399 and *PHO2* transcript abundance profiles following the imposed stress, similarities in the molecular outcome of the stress response of Col-0/NaCl, *MIM399*/NaCl and *MIR399*/NaCl seedlings were evident. Namely, miR399 abundance was elevated in salt-stressed Col-0, *MIM399* and *MIR399* plants resulting in altered *PHO2* transcript abundance (Figure 3B,C). Altered *PHO2* transcript abundance due to the enhancement of the degree of miR399-directed expression regulation would, in turn, relax the degree of PHO2-mediated ubiquitination of its PHT1 PO_4_ transporter proteins at the posttranslational level [48,49,50]. Relaxation of PHO2-mediated posttranslational regulation of PHT1 PO_4_ transporter activity would enhance the degree of Pi acquisition and translocation from the root to the shoot tissue via the xylem [48,49,50]. Enhanced PHT1 activity is best evidenced here by significantly elevated *PHT1;4* and *PHT1;9* transcript abundance in salt-stressed Col-0, *MIM399* and *MIR399* seedlings (Figure 4B,C). Increased Pi availability in the aerial tissues of *Arabidopsis* plants exposed to salt stress would provide *Arabidopsis* with additional cellular resources to respond to this form abiotic stress. More specifically, considering that salt stress is well documented to have a major negative impact on primary plant functions such as photosynthesis and cellular growth in the aerial tissues [52], the increased allocation of Pi, a key building block of cellular energy and component of photosynthesis, in the aerial tissues of a salt-stressed *Arabidopsis* plant, could aid in the attempt of the plant to negate the negative consequences of salt stress, such as combating reductions in its photosynthetic capacity, or to redirect this precious resource to other cellular growth parameters with high chemical energy demands.

## 4. Materials and Methods

### 4.1. Plant Material and Agrobacterium tumefaciens-Mediated Transformation of Arabidopsis thaliana

Seeds sourced from wild-type *Arabidopsis thaliana* (ecotype; Columbia-0 (Col-0)) plants and those of the *MIM399* and *MIR399* transformant lines were surface sterilized for 90 minutes (min) in a sealed chamber with chlorine gas. Post sterilization, seeds were plated out onto plates that contained standard *Arabidopsis* growth medium (half-strength Murashige and Skoog (MS) salts). The plates were sealed with gas permeable tape and incubated at 4 °C in the dark for 48 hours (h) for stratification to ensure uniform germination and subsequent developmental progression of the Col-0, *MIM399* and *MIR399* plant lines. Following the 48 h stratification period, the sealed plates were transferred to a temperature controlled growth cabinet (A1000 Growth Chamber, Conviron, Australia) and cultivated under a 16 h light (100–120 μmol m^−2^ s^−1^) and 8 h dark cycle with a day/night temperature of 22 °C/18 °C for an eight day period. The salt stress treatment regime was conducted as according to [19]. In brief, equal numbers (n = 48) of 8-day-old Col-0, *MIM399* and *MIR399* seedlings were transferred under sterile conditions to either (1) fresh standard *Arabidopsis* growth medium (control treatment), or (2) fresh *Arabidopsis* growth medium that had been supplemented with 150 mM NaCl (salt stress treatment). Post seedling transfer, the control and salt stress treatment plates were returned to the temperature-controlled growth cabinet and were cultivated for an additional 7-day period. All of the phenotypic and molecular assessments reported here were conducted on 15-day-old Col-0, *MIM399* and *MIR399* seedlings following this 7-day growth period.

For the generation of plant material for *Agrobacterium tumefaciens* (*Agrobacterium*)-mediated transformation, Col-0 seedlings that had been germinated and cultivated on standard *Arabidopsis* growth medium for a 14-day period were subsequently transferred to soil (Seeds and Cuttings Mix, Debco, Australia), and maintained under the same growth conditions as outlined above until the plants had progressed to the reproductive stage of development. All secondary inflorescences were trimmed from the Col-0 plants, as were any siliques or open flowers that had formed on the primary inflorescence. The remaining floral material, namely the terminal floral bud, was used for *Agrobacterium*-mediated transformation according to the protocol of [53]. The plant expression vectors used for floral dip transformation of *Arabidopsis* were p*AtMIM399* and p*AtMIR399* and these two plant expression vectors were generated via placing the artificially synthesized (Integrated DNA Technologies, Coralville, IA, USA) *INDUCED BY PHOSPHATE STARVATION1* (*IPS1*; *AT3G09922*) and *PRE-MIR399C* (*AT5G62162*) sequences respectively, behind the *Cauliflower mosaic virus* (*CaMV*) *35S* promoter of the pBART plant expression vector backbone. To screen for putative transformants, the T_1_ seeds harvested from the ”*dipped*” T_0_ Col-0 plants were sterilized, stratified and cultivated exactly as outlined above, however, the *Arabidopsis* growth medium was supplemented with the selective agent, phosphinothricin (PPT), at a concentration of 10 mg/L. Each resistant T_1_ plant was allowed to fully mature on soil to enable the collection of T_2_ seeds. The T_2_ generation was also exposed to the same selective process in addition to the zygosity of each putative transformant line being determined in this generation. Of the T_2_ transformant lines determined to be homozygous for the introduced transgenes, the ”*best performing*” transformant line was selected as the representative plant line of the *MIM399* and *MIR399* populations for subsequent phenotypic and molecular analyses which were conducted on the T_3_ generation.

### 4.2. Phenotypic and Physiological Assessment of Arabidopsis Transformant Lines

In order to compare the *MIM399* and *MIR399* transformant lines with unmodified Col-0 plants, 15-day-old Col-0, *MIM399* and *MIR399* seedlings were phenotypically assessed via the measurement of their fresh weight, rosette area and primary root length. The fresh weight (mg), rosette area (mm^2^) and primary root length (mm) metrics obtained for control grown Col-0 seedlings were converted to a value of 100% for ease of comparison of the same metrics obtained for control grown *MIM399* and *MIR399* plants, and for salt-stressed Col-0, *MIM399* and *MIR399* plants. It is also important to note here that for primary root length measurement, after the transfer of 8-day-old Col-0, *MIM399* and *MIR399* seedlings to fresh control or salt stress media plates, the plates were orientated vertically upon their return to the temperature-controlled growth cabinet for the entire 7-day growth period as we have reported previously [19,20].

The abundance of anthocyanin and chlorophyll *a* and *b* were also quantified in control and salt stressed 15-day-old Col-0, *MIM399* and *MIR399* seedlings. As for the phenotypic measurements, the abundance of anthocyanin (micrograms per gram of fresh weight (μg/g FW)) and of chlorophyll *a* and *b* (milligram per gram of fresh weight (mg/g FW)) obtained for control grown Col-0 seedlings were converted to a value of 100% for ease of comparison of the same metrics obtained for control grown *MIM399* and *MIR399* plants, and for salt-stressed Col-0, *MIM399* and *MIR399* seedlings. Anthocyanin content was determined according to [54]. In brief, 100 mg of freshly ground leaf material was incubated in 1.0 mL of acidic methanol (contained 1.0% *v*/*v* HCl) for 2 h at 4 °C. The ground leaf material was then pelleted by centrifugation at 15,000× *g* for 5 min at room temperature. The absorbance (A) of the resulting supernatant was measured at 530 and 657 nanometers (nm) in a GENESYS 10S spectrophotometer (ThermoFisher Scientific, Australia), and via the use of acidic methanol as the blanking solution. The anthocyanin content of each sample was determined by use of the equation: Anthocyanin (μg/g FW) = (A530 − 0.25 × A657)/sample weight (g). The chlorophyll *a* and *b* content were calculated according to [55]. In brief, 100 milligrams (mg) of ground leaf material were incubated in 1.0 mL of 80% (*v*/*v*) acetone in the dark for 24 h at room temperature. The ground leaf material was pelleted by centrifugation at 15,000× *g* for 5 min at room temperature. The A of the resulting supernatant was measured at 646 and 663 nm in a GENESYS 10S spectrophotometer (ThermoFisher Scientific, Australia), and via the use of 80% acetone as the blanking solution. Next, the chlorophyll *a* and *b* content of each sample was determined using the Lichtenthaler’s equations exactly as outlined in [55], and these initially determined values were subsequently converted to micrograms per gram of fresh weight (μg/g FW).

### 4.3. Quantification of MicroRNA Abundance and of Gene Expression Via the Use of the Quantitative Reverse Transcriptase Polymerase Chain Reaction Approach

For each molecular assessment reported in this study, total RNA was extracted from four biological replicates (each biological replicate contained tissue sampled from eight individual plants) of 15-day old control grown and salt stressed Col-0, *MIM399* and *MIR399* seedlings using TRIzol^TM^ Reagent according to the manufacturer’s (Invitrogen^TM^, Waltham, MA, USA) instructions. The quality of the extracted total RNA was visually assessed via a standard electrophoresis approach on a 1.2% (*w*/*v*) ethidium bromide stained agarose gel. For each high quality total RNA preparation, the quantity of the total RNA extracted was next determined using a NanoDrop spectrophotometer (NanoDrop^®^ ND-1000, Thermo Scientific, Waltham, MA, USA). The synthesis of a miR399-specific complementary DNA (cDNA) was conducted as previously reported in [19,47] using a protocol adapted from [56]. In brief, 200 nanograms (ng) of total RNA was treated with 0.2 units (U) of DNase I according to the manufacturer’s (New England Biolabs, Ipswich, MA, USA) instructions. The DNase I-treated total RNA was next used as template for cDNA synthesis along with 1.0 U of ProtoScript^®^ II Reverse Transcriptase (New England Biolabs, Ipswich, MA, USA) and the cycling conditions of 1 cycle of 16 °C for 30 min; 60 cycles of 30 °C for 30 s, 42 °C for 30 s, and 50 °C for 2 s, and 1 cycle of 85 °C for 5 min. A global, high molecular weight cDNA library for gene expression quantification was constructed as according to [57,58]. Specifically, 5.0 μg of total RNA was initially treated with 5.0 U of DNase I according to the manufacturer’s protocol (New England Biolabs, Ipswich, MA, USA). The DNase I-treated total RNA was next purified using a RNeasy Mini Kit (Qiagen, Australia) and 1.0 μg of this preparation was used as template for cDNA synthesis along with 1.0 U of ProtoScript^®^ II Reverse Transcriptase (New England Biolabs,Ipswich, MA, USA) and 2.5 mM of oligo dT_(18)_, according to the manufacturer’s instructions. All generated, single-stranded cDNAs were next diluted to a working concentration of 50 ng/μL in RNase-free water prior to their use as a template for the quantification of the abundance of either miR399 or of gene transcripts. In addition, all RT-qPCRs used the same cycling conditions of 1 cycle of 95 °C for 10 min, followed by 45 cycles of 95 °C for 10 s and 60 °C for 15 s, and the GoTaq^®^ qPCR Master Mix (Promega, Australia) was used as the fluorescent reagent for all performed RT-qPCR experiments. miR399 abundance and gene transcript expression was quantified using the 2^−ΔΔCT^ method with the small nucleolar RNA, *snoR101*, and *UBIQUITIN10* (*UBI10*; *AT4G05320*), used as the respective internal controls to normalize the relative abundance of each assessed transcript. The sequence of each DNA oligonucleotide used in this study either for the synthesis of a miR399-specific cDNA, or to quantify transcript abundance via RT-qPCR is provided in Appendix A.

## 5. Conclusions

Via the phenotypic, physiological and molecular assessment of wild-type *Arabidopsis* (Col-0) plants, and the *MIM399* and *MIR399* transformants, plant lines molecularly modified to have altered miR399 abundance, here we show that the PO_4_ responsive miRNA, miR399, is also responsive to salt stress. Interestingly, at the phenotypic and physiological levels, both the *MIM399* and *MIR399* transformant lines appeared to be more sensitive to the imposed stress regime than did unmodified wild-type *Arabidopsis* plants. However, in spite of the phenotypic and physiological differences displayed by 15-day-old Col-0, *MIM399* and *MIR399* seedlings following their cultivation for a 7-day period in the presence of 150 mM NaCl, a shared molecular outcome was observed. Namely, elevated miR399 abundance and dysfunctional miR399-directed regulation of *PHO2* target gene expression was readily evident in salt-stressed Col-0, *MIM399* and *MIR399* plants. Dysfunctional miR399-directed regulation of *PHO2* gene expression, and therefore PHO2 protein activity, was evidenced via elevated expression of either *PHT1;4* and *PHT1;9* in salt-stressed Col-0, *MIM399* and *MIR399* plants; two transcripts that encode for PO_4_ transporter proteins which form PHO2 regulated targets at the posttranslational level in *Arabidopsis*. This finding strongly suggests that relaxation of PHO2-mediated regulation of PHT1 PO_4_ transporter protein activity would enhance the degree of Pi acquisition by the roots and subsequent translocation to the shoot tissue via the xylem when *Arabidopsis* is cultivated in the presence of NaCl. An increased allocation of Pi, a key building block of cellular energy and component of photosynthesis, in the aerial tissues of a salt-stressed *Arabidopsis* plant, could aid in the attempt of the plant to negate the negative consequences of salt stress, such as combating reductions in its photosynthetic capacity, or to redirect this precious resource to other cellular growth parameters or adaptive responses that demand high amounts of chemical energy.

## Figures and Tables

**Figure 1 plants-10-00073-f001:**
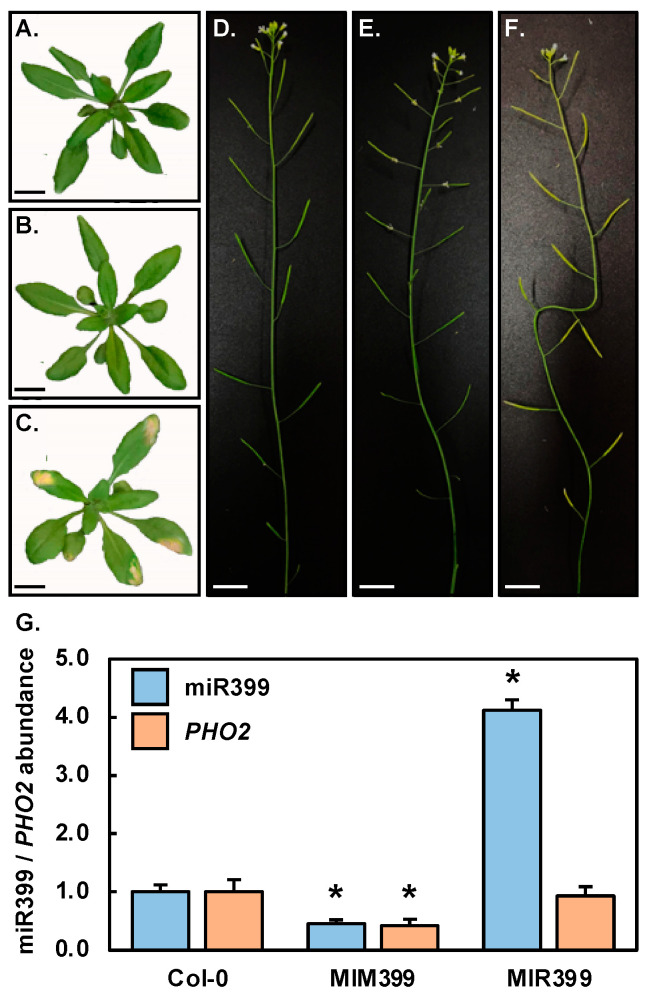
*Arabidopsis* transformant lines, *MIM399* and *MIR399* plants, molecularly modified to have altered miR399 abundance. Vegetative phenotypes displayed by 25-day-old Col-0 (**A**), *MIM399* (**B**) and *MIR399* (**C**) plants. (**A**–**C**) Scale bar = 1.0 cm. The primary inflorescence and siliques of 40-day-old Col-0 (**D**), *MIM399* (**E**) and *MIR399* (**F**) plants. (**D**–**F**) Scale bar = 2.0 cm. (**G**) RT-qPCR quantification of miR399 abundance and the expression of its target gene, *PHO2*, in the rosette leaves of 25-day-old Col-0, *MIM399* and *MIR399* plants. Error bars represent the standard deviation of four biological replicates and the presence of an asterisk above a column represents a statistically significant difference in the abundance of either miR399 or its *PHO2* target transcript between *MIM399*, *MIR399* and Col-0 plants (*p*-value: * < 0.05).

**Figure 2 plants-10-00073-f002:**
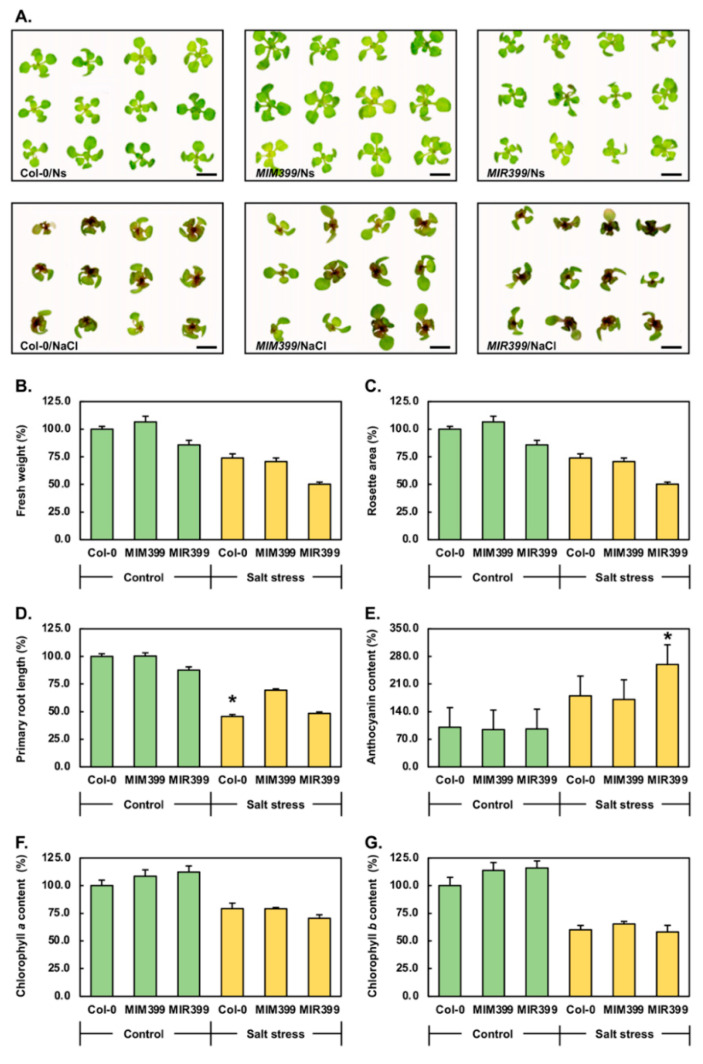
Phenotypic and physiological responses of Col-0, *MIM399* and *MIR399* plants to salt stress. (**A**) Phenotypes displayed by 15-day-old Col-0/Ns, *MIM399*/Ns, *MIR399*/Ns, Col-0/NaCl, *MIM399*/NaCl and *MIR399*/NaCl seedlings. Quantified phenotypic parameters of fresh weight (**B**), rosette area (**C**) and primary root length (**D**) for 15-day-old Col-0/Ns, *MIM399*/Ns, *MIR399*/Ns, Col-0/NaCl, *MIM399*/NaCl and *MIR399*/NaCl seedlings. Quantification of the physiological parameters of anthocyanin (**E**), chlorophyll *a* (**F**) and chlorophyll *b* (**G**) content in 15-day-old Col-0/Ns, *MIM399*/Ns, *MIR399*/Ns, Col-0/NaCl, *MIM399*/NaCl and *MIR399*/NaCl seedlings. Error bars represent the standard deviation of four biological replicates and the presence of an asterisk above a column represents a statistically significant difference in either a phenotypic or physiological parameter between the Col-0 plants and the *MIM399* and *MIR399* transformant lines (*p*-value: * < 0.05).

**Figure 3 plants-10-00073-f003:**
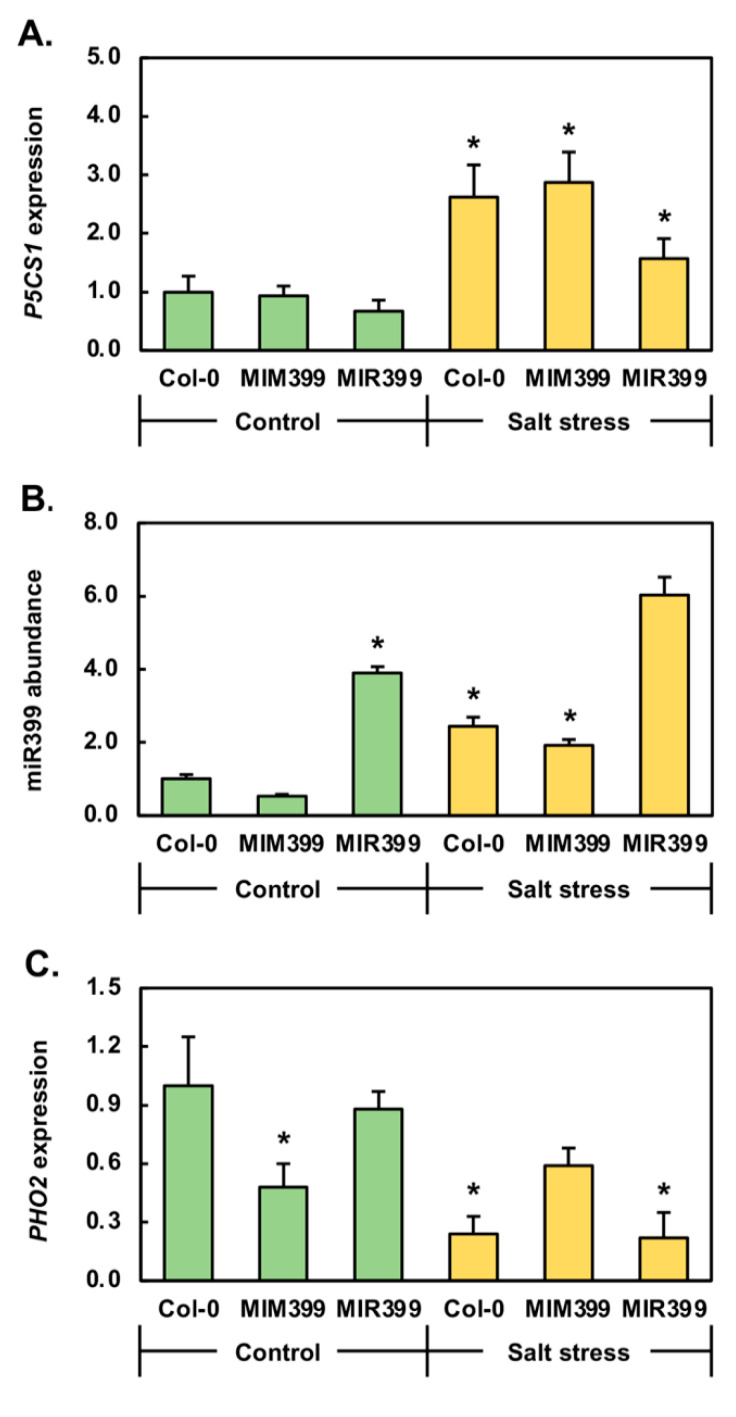
RT-qPCR analysis of *P5CS1*, miR399 and *PHO2* transcript abundance in 15-day-old control and salt-stressed Col-0, *MIM399* and *MIR399* seedlings. Quantification of *P5CS1* (**A**), miR399 (**B**) and *PHO2* (**C**) expression via RT-qPCR assessment of 15-day-old control and salt-stressed Col-0, *MIM399* and *MIR399* seedlings. Error bars represent the standard deviation of four biological replicates and the presence of an asterisk above a column represents a statistically significant difference in the abundance of an assessed transcript between the Col-0, *MIM399* and *MIR399* plant lines (*p*-value: * < 0.05).

**Figure 4 plants-10-00073-f004:**
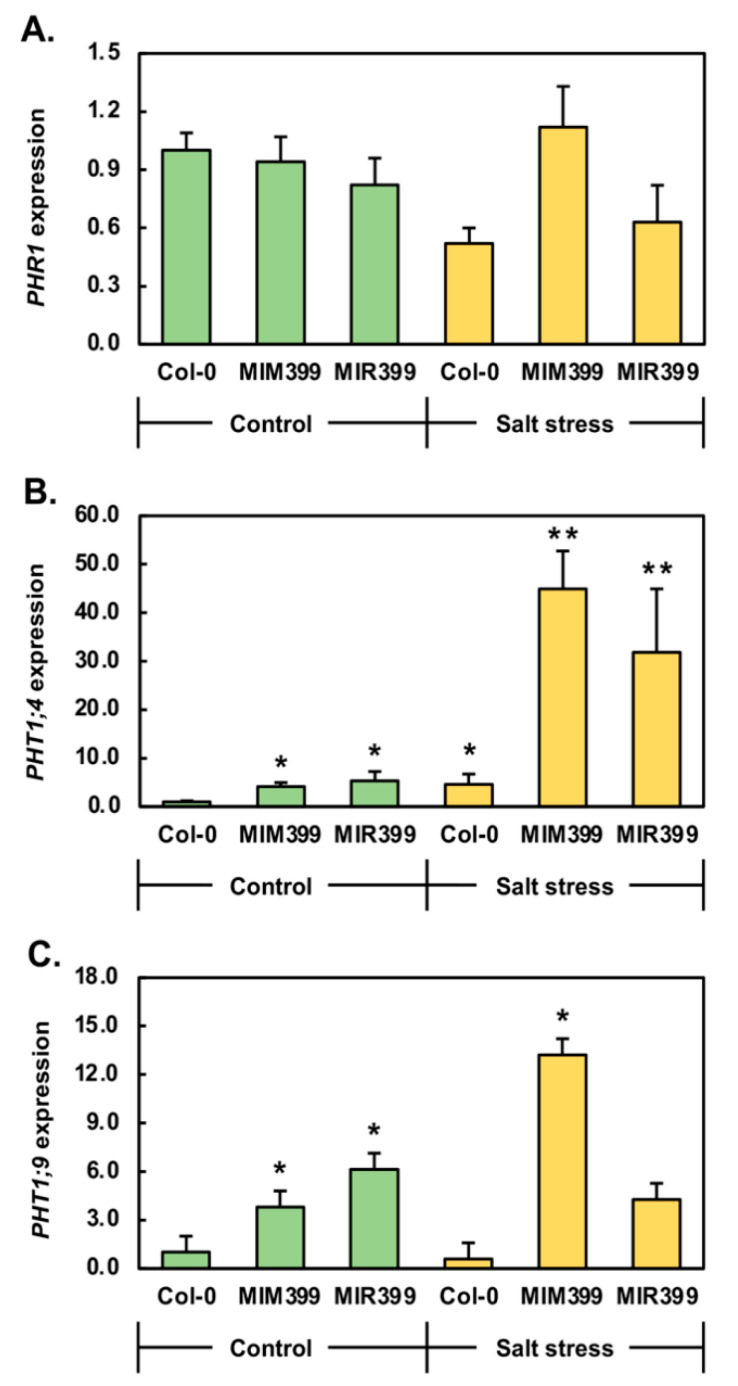
Quantification of *PHR1*, *PHT1;4* and *PHT1;9* expression via RT-qPCR analysis of control and salt-stressed Col-0, *MIM399* and *MIR399* 15-day-old *Arabidopsis* seedlings. Assessment of the expression of *PHR1* (**A**), *PHT1;4* (**B**) and *PHT1;9* (**C**) in 15-day-old control and salt stressed *Arabidopsis* plant lines, Col-0, *MIM399* and *MIR399*. Error bars represent the standard deviation of four biological replicates and the presence of an asterisk above a column represents a statistically significantly difference in the abundance of an assessed transcript between the compared Col-0, *MIM399* and *MIR399* plant lines (*p*-value: * < 0.05; ** < 0.005).

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
