# Peer review of "Molecular Manipulation of the miR399/PHO2 Expression Module Alters the Salt Stress Response of Arabidopsis thaliana"

_plants, 2020, doi:10.3390/plants10010073_

Round 1

Reviewer 1 Report

Dear Authors of manuscript entitled “Molecular manipulation of the miR399/PHO2 expression module alters the salt stress response of Arabidopsis thaliana”, below please find my comments and suggestions:

In my opinion Introduction is very well described. This section presents all necessary issues, which are important to understand the background of undertaken research.

The methods were well presented, the performance of the experiments is well described and allows for their repetition. However, I suggest that authors should cite the source publications which used the methods described. It seems to me that there were more than just the three mentioned. If the authors used their own procedures, or have already revised ready-made procedures, please mention this as well.

I have to admit that the authors presented the obtained results efficiently, which was not so obvious with such a large scope of the experiments carried out. I have read this article with great interest, and I believe that the way in which the results are discussed is correct and helps the reader follow this results.

The discussion does not raise my objections.

However, taking into account the scope of the studies performed and the number of results obtained, it might be appropriate to include a short summary and the most important conclusions at the end of the manuscript.

I believe that the presented results will have many readers and the article will provide valuable knowledge to other scientists.

Author Response

Dear Reviewer,

The authorship team thank you kindly for your thorough and highly helpful review of our originally submitted manuscript. In the revised version of the manuscript we have addressed each of the concerns that you raised in your review. Please find our responses to each of the issues that you raised in your review below, and again, we thank you for taking the time to provide a number of helpful suggestions that will improve the quality of our study in its revised version.

Regards,

Andrew (on behalf of the authorship team).

******************************************************************

Dear Authors of manuscript entitled “Molecular manipulation of the miR399/PHO2 expression module alters the salt stress response of Arabidopsis thaliana”, below please find my comments and suggestions:

In my opinion Introduction is very well described. This section presents all necessary issues, which are important to understand the background of undertaken research.

**The authors thank the Reviewer for this kind comment.

The methods were well presented, the performance of the experiments is well described and allows for their repetition. However, I suggest that authors should cite the source publications which used the methods described. It seems to me that there were more than just the three mentioned. If the authors used their own procedures, or have already revised ready-made procedures, please mention this as well.

**The authors again thank the Reviewer for this helpful suggestion and we do apologise for the omission of this information in the original version of our manuscript. We have now added additional citations to the Materials and Methods section of the Revised manuscript in order to provide additional information and resources for other researchers wishing to perform similar analyses in the future.

I have to admit that the authors presented the obtained results efficiently, which was not so obvious with such a large scope of the experiments carried out. I have read this article with great interest, and I believe that the way in which the results are discussed is correct and helps the reader follow this results.

**Thank you to the Reviewer for this positive comment.

The discussion does not raise my objections.

**Thank you kindly to the Reviewer for this positive comment.

However, taking into account the scope of the studies performed and the number of results obtained, it might be appropriate to include a short summary and the most important conclusions at the end of the manuscript.

**We thank the Reviewer for this helpful and insightful suggestion. We have now added a Conclusion section to the revised version of our manuscript which highlights the major findings of our study. We do apologise for its omission in the original version of our manuscript.

I believe that the presented results will have many readers and the article will provide valuable knowledge to other scientists

**The authors are highly appreciative of this kind comment from the Reviewer.

Reviewer 2 Report

Dear Authors

The present manuscript "Molecular Manipulation of the miR399/PHO2 Expression Module Alters the Salt Stress Response of Arabidopsis thaliana" demonstrated the molecular basis of essential biological processes to mount an adaptive response to salt stress. The study have been well planned in reference to experimental design. Introduction and methods are well explained and discussed nicely. Although, there is a need to draw a clear concluding remark in order to communicate the key findings. The language may be improved a little bit as well.

Thank you

Author Response

Dear Reviewer,

On behalf of the authorship team I wish to thank you for taking the time to review our submitted study and for providing a number of helpful suggestions for the manuscript's improvement in its revised version. Please find our response to your review below.

Kind Regards,

Andrew

*****************************************************************Dear Authors

The present manuscript "Molecular Manipulation of the miR399/PHO2 Expression Module Alters the Salt Stress Response of Arabidopsis thaliana" demonstrated the molecular basis of essential biological processes to mount an adaptive response to salt stress. The study have been well planned in reference to experimental design. Introduction and methods are well explained and discussed nicely. Although, there is a need to draw a clear concluding remark in order to communicate the key findings. The language may be improved a little bit as well.

**We have now added a Conclusions section to the revised version of our manuscript which highlights the major findings of our study as well as to provide possible biological reasoning for our reported findings. We do thank the Reviewer for making this helpful suggestion, a suggestion which the authors believe greatly adds to the impact of our revised manuscript and the results it reports.

**We have also added additional citations to the Materials and Methods section of the revised manuscript in order to provide greater detail on each of the experiments reported on in our study in order to assist other researchers wishing to perform similar experimentation.

**We have edited the text of the manuscript throughout all sections of the revised version of the manuscript in an attempt to improve the language of the manuscript. The authors thank the Reviewer for identifying this issue in their review of our original manuscript.